# Tinnitus and occupational noise exposure among informal generator technicians in Nigeria: A pilot cohort study

Fatimah Isma'il Tsiga-Ahmed[1], Nafisatu Bello-Muhammad[2], Abdulazeez Ahmed[2]*

1 Department of Community Medicine, Bayero University & Aminu Kano Teaching Hospital, Kano, Nigeria,
2 Department of Otolaryngology, Bayero University & Aminu Kano Teaching Hospital, Kano, Nigeria

* aoahmad.oto@buk.edu.ng

## Abstract

Occupational noise exposure is a major cause of auditory dysfunction worldwide, and generator technicians in Nigeria represent a vulnerable informal workforce with prolonged exposure to high decibel noise. We conducted a population-based, cross-sectional cohort study of 73 generator technicians (age range 14–57 years, mean 34.9 ± 10.2 years). Subjective, non-pulsatile tinnitus was assessed via a structured questionnaire, audiometric thresholds were measured using pure tone audiometry (PTA), and cochlear function was evaluated with distortion product otoacoustic emissions (DPOAEs). Logistic regression was used to identify independent predictors of tinnitus. Overall tinnitus prevalence was 45.2%, increasing across age groups from 33.3% in participants aged 14–24 years to 60% in those aged ≥55 years. Cross-tabulation revealed tinnitus was most common among participants with normal PTA but failed OAE (100%). Logistic regression identified ≥10 years of occupational noise exposure (adjusted odds ratio [aOR] = 4.44) and OAE failure (aOR = 10.1) as independent predictors. Tinnitus was highly prevalent among this cohort of generator technicians and strongly associated with prolonged exposure and OAE failure. These findings underscore the complementary diagnostic role of OAE testing and highlight the urgent need for workplace hearing conservation strategies in informal sectors.

## Introduction

Tinnitus, typically described as a subjective, non-pulsatile perception of sound in the absence of an external stimulus, affects an estimated 4.6% to 43% of individuals from Asia, North Africa to the United States, with lower estimates in community samples and higher rates in older and/or noise-exposed cohorts [1–6]. Though its aetiology remains multifactorial, it is widely recognized as a common symptom in populations exposed to prolonged community, leisure or occupational noise [7–9]. Beyond its auditory manifestations, persistent tinnitus can negatively impact sleep, emotional

**Data availability statement:** All relevant data are within the paper and its Supporting Information files.

**Funding:** The authors received no specific funding for this work.

**Competing interests:** The authors have declared that no competing interests exist.

well-being, and overall quality of life, with links to anxiety and depression increasingly being reported [10–12].

In occupational contexts, tinnitus may be an early marker of cochlear dysfunction, often preceding measurable hearing loss [13,14]. While industrial workers in regulated environments are frequently studied, informal-sector workers, such as generator technicians, welders, and artisans, operate in uncontrolled acoustic conditions, frequently exceeding 90 dB(A) without adequate protection or access to routine hearing screening [15–19]. Despite this heightened vulnerability, the auditory health of informal workers remains severely underexplored, particularly in low-resource urban economies.

In Nigeria, unreliable power supply has spurred widespread reliance on electricity generators, resulting in the proliferation of informal generator-repair businesses concentrated in urban centres [18,20]. Workers in this setting are regularly exposed to prolonged and intense noise, yet there is limited research on their auditory profiles, especially early symptoms like tinnitus that may reflect subclinical cochlear changes.

Furthermore, emerging literature suggests that tinnitus is shaped not only by noise exposure but also by sociodemographic and psychological factors, including age, education, emotional stress, and beliefs about hearing health [21–23]. Understanding the intersection of these factors in informal noise-exposed populations could inform accessible screening tools and targeted preventive strategies, particularly where conventional diagnostics like audiometry and otoacoustic emissions (OAE) testing are seldom available.

This cohort study aims to assess the prevalence and predictors of tinnitus among informal generator technicians in Northern Nigeria. By examining sociodemographic, occupational, and audiological variables, including pure-tone audiometry and OAE, we seek to identify markers of early auditory dysfunction and highlight opportunities for community-based hearing conservation in underserved work environments.

## Materials and methods

### Ethics statement

Ethical approval for this study was obtained from the Kano State Ministry of Health Ethics Committee (NHREC/17/03/2018) and the Bayero University Health Research Ethics Committee (NHREC/BUK-NHREC/348/10/2311). Written informed consent was obtained from all participants (including adolescent assent obtained for minors) prior to enrolment. The study was conducted in accordance with the ethical principles outlined in the Declaration of Helsinki, as revised in 2013.

### Study design and setting

This was a community-based pilot cross-sectional cohort study conducted in Kano State, Northwestern Nigeria. Data collection spanned a four-month period (4th June – 30th September, 2024), targeting densely populated metropolitan areas characterized by informal electrical generator repair workshops.

## Study population and eligibility criteria

All participants were male, reflecting the gender composition of informal generator technicians in Nigeria. This occupational demographic is shaped by cultural and societal norms, and the sample was representative of the field population. Seventy-three generator technicians aged 14–57 years, randomly selected from union registries across four local government areas. Written informed consent was obtained from all participants including assent from minors/guardians. Individuals previously diagnosed with otologic disease or those older than 57 years (to minimize effect of confounding from age-related hearing loss) were excluded.

## Sampling procedure

This pilot cohort study recruited 73 informal generator technicians. The initial target sample size was based on an assumed tinnitus prevalence of 20% in noise-exposed populations, consistent with previous occupational studies reporting prevalence estimates in the range of 20–30% [24,25]. With a desired precision of ±10% at the 95% confidence level, the required sample size was approximately 62. To account for potential non-response, this was increased by 10% to 68. Ultimately, 73 participants were recruited.

A two-stage random sampling strategy was applied:

**Stage 1:** Four (4) metropolitan Local Government Areas (LGAs), Nassarawa, Tarauni, Fagge, and Kumbotso, were randomly selected out of a total of eight to enhance reach and feasibility, given that generator technicians in Kano are primarily concentrated in the urban areas.

**Stage 2:** From the union registries obtained in each selected LGA, generator technicians were randomly selected using computer-generated numbers. The number of participants from each LGA was proportionally allocated based on registry size (~250 members).

## Data collection tools and measures

A structured interviewer-administered, pretested questionnaire was used to collect data on sociodemographic variables (age, marital status, education, income), occupational exposure (years of experience, daily noise exposure), hearing loss, family history of deafness (within immediate relatives, genetic/acquired deafness) and tinnitus history as well as PTA, OAE outcome. Tinnitus was defined as self-reported perception of ringing, buzzing, or other phantom auditory sensations in one or both ears.

**Tinnitus measurement.** Participants were asked whether they experienced ringing, buzzing, or other phantom auditory sensations. Validated instruments were not used because Hausa-language versions were not available, and adaptation was beyond the scope of this study.

## Hearing assessment procedures

All assessments were conducted by trained and certified Audiometricians with >5 years' experience. Otoscopy was performed first to identify external or middle ear conditions.

**Pure-Tone Audiometry (PTA).** Conducted in a sound-treated booth using a calibrated diagnostic audiometer (Model: Interacoustic AT235H). Audiometric thresholds were measured, across 0.5–8 kHz frequencies. Impairment was defined as PTA > 25 dB HL.

**Otoacoustic Emissions (OAE) testing.** Performed using a portable screener (Sentiero-Path Medical) after PTA. Bilateral testing was conducted, and results were categorized as "pass" only when both ears passed. Any unilateral or bilateral failure was considered indicative of cochlear dysfunction (OAE Fail). Otoacoustic Emissions testing 'pass' was defined as a signal-to-noise ratio ≥6 dB at ≥3 frequencies; while 'fail' was below this threshold.

Ambient noise levels at the generator workshops were measured prior to testing using a calibrated sound level meter (SL-4010, Lutron Electronics), and ranged from 95–120 dB(A). Given these high on-site noise levels, all audiometric and OAE assessments were conducted the following morning after each workday at our health facility, allowing outer hair cell rest and recovery before testing and ensuring an environment that met recommended ambient noise standards for hearing testing.

**Rationale for dual hearing assessment.** Both PTA and OAE were employed due to their distinct but complementary pathways. While PTA assesses behavioural hearing thresholds, OAEs provide objective insight into outer hair cell function. Several studies have highlighted the enhanced sensitivity and versatility of OAE testing, especially in detecting subclinical cochlear changes that may precede audiometric loss [26–28]. Their combined use strengthens diagnostic reliability and enables a broader characterization of auditory health.

## Data analysis

Data were entered into Microsoft Excel and analysed using SPSS version 20 (IBM Corp., Armonk, NY). Frequencies and percentages summarized categorical variables, while continuous data were reported as means ± SD or medians with interquartile ranges depending on distribution (assessed via Kolmogorov-Smirnov test). Chi-square or Fisher's exact tests assessed bivariate relationships, with independent t-tests or Mann–Whitney U tests for continuous variables. Variables with $p < 0.10$ were entered into logistic regression to identify independent predictors of tinnitus. Adjusted odds ratios (aOR) with 95% confidence intervals were reported, and a p-value $< 0.05$ was considered statistically significant. Age was explored using decade-based categories and visualised through prevalence tables and a LOESS curve.

## Community/Participants engagement

In addition to obtaining informed consent, brief counselling and health education were provided to all participants to ensure understanding of the study purpose, procedures, and potential benefits. As part of routine engagement, we delivered short health-talks on safe listening practices, risks of chronic noise exposure, and simple strategies for ear care appropriate for informal work settings. After the assessments, each participant received confidential, individualized feedback on his audiometric and OAE results and was advised on next steps where abnormalities were identified. Although we did not provide physical hearing protection devices during this pilot phase, participants were counselled on the importance of hearing protection and methods to reduce daily exposure within the constraints of their working environment.

## Results

### Demographics

A total of 73 generator-exposed workers were ultimately enrolled for this study. Among this cohort, 33 (45.2%) reported tinnitus. The mean age was 34.9 ± 10.2 years. Most participants (64.4%) were aged 26–45 years, 68.5% were married, and 71.2% had at least 10 years of occupational experience. Over 70% reported daily exposure to generator noise exceeding 8 hours (Table 1). All the participants were male and reported no use of hearing protection device.

### Age-related distribution of tinnitus

To explore potential age-related trends, a table of tinnitus prevalence across decade-based age groups was constructed (Table 2) and a LOESS curve of prevalence versus age (S1 Fig) generated. The smoothed curve shows a gradual upward pattern with advancing age, suggesting cumulative exposure-related vulnerability. In logistic regression scaled per 10-year increase in age, the association remained positive but not statistically significant, likely reflecting the small sample size and collinearity between age and years in occupation in this pilot cohort.

**Table 1. Demographic and occupational characteristics of generator technicians (N = 73).**

| Variable | Category | n (%) |
|---|---|---|
| **Age Group (years)** | 14–24 | 9 (12.3) |
| | 25–34 | 28 (38.4) |
| | 35–44 | 19 (26.0) |
| | 45–54 | 12 (16.4) |
| | ≥55 | 5 (6.8) |
| **Marital Status** | Married | 50 (68.5) |
| | Single | 23 (31.5) |
| **Education Level** | Primary | 20 (27.4) |
| | Secondary | 41 (56.2) |
| | Tertiary | 2 (2.7) |
| | Non-literate | 10 (13.7) |
| **Household Income (₦/month)** | ≤5,000 | 36 (49.3) |
| | 6,000–10,000 | 21 (28.8) |
| | 11,000–20,000 | 7 (9.6) |
| | 21,000–30,000 | 2 (2.7) |
| | ≥31,000 | 0 (0.0) |
| | Prefer not to say | 7 (9.6) |
| **Ethnicity** | Hausa | 61 (83.6) |
| | Yoruba | 4 (5.5) |
| | Igbo | 0 (0.0) |
| | Others | 8 (11.0) |
| **Years of Exposure** | <10 | 21 (28.8) |
| | ≥10 | 52 (71.2) |
| **Daily Hours of Exposure** | 6–8 | 17 (23.3) |
| | 9–11 | 46 (63.0) |
| | ≥12 | 10 (13.7) |
| **Family History of Hearing Loss** | Yes | 11 (15.1) |
| | No | (84.9) |

*Footnotes: Audiometric impairment defined as PTA > 25 dB HL. Family history refers to immediate relatives, self-reported. Any unilateral or bilateral failure was considered indicative of cochlear dysfunction (OAE Fail).*

**Table 2. Prevalence of self-reported tinnitus by age group.**

| Age group (years) | Tinnitus cases | Prevalence (%) | 95% CI (%) | Total |
|---|---|---|---|---|
| 14-24 | 3 | 33.3 | 7.5 - 70.1 | 9 |
| 25-34 | 12 | 42.9 | 24.5 - 62.8 | 28 |
| 35-44 | 9 | 47.4 | 24.4 - 71.1 | 19 |
| 45-54 | 6 | 50.0 | 21.1 - 78.9 | 12 |
| ≥55 | 3 | 60.0 | 14.7 - 94.7 | 5 |
| **Total** | 33 | 45.2 | 33.2 - 57.6 | 73 |

## Audiological relationships

A cross-tabulation of PTA and OAE results by tinnitus status is shown in Table 3. Of note, Tinnitus prevalence was highest among participants with normal PTA but failed OAE (100% [95% CI: 54.1–100.0]), underscoring the diagnostic value of subclinical cochlear testing. All participants with OAE pass had lower prevalences (37.5% [95% CI: 18.8–59.4] & 24.0% [95% CI: 9.4–45.1]) respectively (Table 3).

## Multivariable analysis

Bivariate analysis revealed statistically significant associations between tinnitus and marital status ($\chi^2 = 14.023$, p < 0.001), years of occupational exposure ($\chi^2 = 9.018$, p = 0.003), audiometric impairment ($\chi^2 = 4.942$, p = 0.026), and OAE failure ($\chi^2 = 14.813$, p < 0.001). The crude association between marital status and tinnitus was eliminated after adjustment for age and occupational exposure, indicating that the initial finding was attributable to confounding rather than a true independent effect. As such, marital status was not considered a meaningful explanatory variable in this study. In the logistic regression model, two factors emerged as statistically significant predictors of tinnitus. First, technicians with ≥10 years of occupational exposure were over four times more likely to report tinnitus compared to their less-exposed counterparts (aOR = 4.44; 95% CI: 1.04–18.95; p = 0.044). Secondly, OAE failure was associated with a ten-fold increase in odds of tinnitus (aOR = 10.1, 95% CI 2.01–48.1; p = 0.004), signifying possible outer hair cell damage or dysfunction (Table 4, S2 Fig). Age, education, audiometric status, daily noise exposure and family history were not significant in the final model. The model explained 33% of the variance (Nagelkerke $R^2 = 0.33$), with an overall predictive accuracy of 72.6% and acceptable calibration (Hosmer–Lemeshow p = 0.064).

## Discussion

This community-based pilot cohort study demonstrated a high tinnitus prevalence (45.2%) among informal generator technicians in metropolitan Northern Nigeria, an occupational cohort often neglected in hearing health research and policy discussions. Most participants were married (68.5%), had ≥10 years of generator-repair experience (53.4%), and reported

**Table 3. Cross-tabulation of PTA and OAE outcomes by Tinnitus Status.**

| PTA×OAE Outcome | Tinnitus Present | Tinnitus Absent | % with Tinnitus | 95% CI (%) | Total |
|---|---|---|---|---|---|
| PTA Impaired + OAE Fail | 12 | 6 | 66.7 | 41.0 - 86.7 | 18 |
| PTA Impaired + OAE Pass | 9 | 15 | 37.5 | 18.8 - 59.4 | 24 |
| PTA Normal + OAE Fail | 6 | 0 | 100.0 | 54.1 - 100.0 | 6 |
| PTA Normal + OAE Pass | 6 | 19 | 24.0 | 9.4 - 45.1 | 25 |
| Total | 33 | 40 | 45.2 | 33.2 - 57.6 | 73 |

**Table 4. Multivariable logistic regression for predictors of Tinnitus.**

| Predictor | (aOR) | 95% CI | p-value |
|---|---|---|---|
| Years of exposure ≥10 | 4.44 | 1.04 – 18.95 | 0.044 |
| OAE Fail (vs Pass) | 10.1 | 2.1 – 48.1 | 0.004 |
| Age (per 10-year increase) | 1.25 | 0.85 – 1.82 | 0.21 |
| Audiometric impairment (PTA > 25 dB HL) | 1.65 | 0.62 – 4.39 | 0.31 |
| Married (vs Single) | 1.40 | 0.52 – 3.75 | 0.49 |

aOR: Adjusted Odds Ratio.

daily noise exposure exceeding 8 hours (93.2%). Tinnitus was strongly associated with prolonged exposure and OAE failure. This finding underscores the urgent need to reframe tinnitus not just as an auditory complaint, but as a sentinel symptom of deeper population-level vulnerabilities related to unregulated noise exposure.

The strong association between prolonged occupational duration (≥10 years) and reported tinnitus points to the cumulative risk posed by chronic exposure to high-decibel generator noise. Such exposure, routinely exceeding 90 dB(A), has been documented in informal trades across Nigeria and sub-Saharan Africa [17,19]. These workers typically lack access to protective equipment, routine hearing screening, and noise hazard education, which places them in what may be described as "high-risk low-protection zones" within urban economies. Our findings support calls for formal recognition of informal workers in national occupational safety frameworks, especially given the scale and importance of the generator repair sector in powering local commerce or small and medium scale enterprises.

Notably, OAE test failure emerged as a significant independent correlate of tinnitus. Table 4 further illustrate that tinnitus can occur with normal audiometric thresholds but failed OAE results. OAE appears to offer greater sensitivity than conventional pure-tone audiometry in detecting early-stage auditory dysfunction, even in those without overt hearing loss [29,30]. This is clinically relevant as it suggests that OAE screening could serve as a cost-effective, field-friendly tool for identifying subclinical cochlear damage in settings where access to audiological infrastructure is limited or absent. Integrating portable OAE devices into community health outreach or primary care screening initiatives may help bridge this diagnostic gap for underserved noise-exposed populations.

Interestingly, audiometric impairment did not retain significance in multivariable analysis. This may reflect the well-documented phenomenon where tinnitus can occur independently of hearing threshold shifts [31,32], suggesting a disconnect between subjective auditory perception and measurable hearing loss. Several studies have also emphasized the role of central auditory plasticity and neural hyperactivity in the generation of tinnitus, even in individuals with clinically normal audiograms [7,32–34]. Other studies emphasize the primacy of peripheral cochlear dysfunction and caution that central hyperactivity is not consistently observed across all patients [35–37].

Though age was not significantly associated with tinnitus in the adjusted models, the distribution in Table 3 reveals symptom presence even among adolescents and younger adults. Several epidemiological data and audiological findings have highlighted that tinnitus prevalence is not restricted to older adults; younger populations exposed to recreational or occupational noise also show measurable rates [38–40]. This underscores the need for early occupational health education targeting youth in informal trades. Furthermore, the absence of a statistically significant age effect in our regression likely reflects the limitations inherent in this small pilot cohort and the strong collinearity between age and years in occupation. Many participants entered generator repair work during adolescence, causing age and cumulative exposure to overlap closely. In such settings, duration of exposure may overshadow chronological age as the more proximal determinant of tinnitus symptoms.

From a public health perspective, our study's findings also hold broader implications. Tinnitus is increasingly linked with mental health disorders, including sleep disturbance, anxiety, and depression [10–12,41]. Informal sector workers, many of whom face economic challenges, occupational instability, and environmental hazards, may be at heightened risk for psychological distress exacerbated by persistent tinnitus. This also calls for integrated care models in primary health systems, where auditory screening is coupled with mental health triage in resource-constrained settings.

Another important yet under-discussed finding is the presence of participants as young as 14 years. Early entry into generator-repair work exposes adolescents to harmful acoustic environments at a critical developmental stage. This observation is supported by some epidemiological studies showing tinnitus prevalence among younger populations, including college-aged adults [39] and even those with clinically normal audiograms [42]. Public health guidance from the American Academy of Pediatrics emphasizes that excessive noise exposure in children and adolescents could cause irreversible auditory damage [43]. Similarly, about a decade ago, the World Health Organization had identified occupational noise as a major contributor to global hearing impairment [44], and the International Labour Organization highlights noise as a workplace hazard, raising ethical concerns when minors are exposed [45]. These findings underscore the ethical and regulatory

questions surrounding child labour in hazardous trades. Preventive health campaigns tailored to younger workers could deliver long-term dividends in reducing lifetime auditory impairment [46]. The economic ripple effects of hearing loss, including reduced productivity, diminished communication efficiency, and increased healthcare costs, are well-documented [47]. In skill-based vocations like generator repair, auditory acuity is not just biologically significant, it is economically essential. Thus, untreated tinnitus and progressive hearing loss could threaten the livelihood of thousands of informal workers.

A key strength of this study is its focus on an understudied, high-risk occupational group in Nigeria's informal sector, where generator technicians are chronically exposed to unregulated noise. The use of both PTA and OAE testing, allowing detection of subclinical dysfunction that might otherwise be missed and the inclusion of sociodemographic and exposure variables all enhance the interpretability of findings. However, several limitations must be acknowledged. Sample size and statistical power: With only 73 participants, the study was adequate for a pilot but lacks power to detect subtler associations, precludes detailed subgroup analyses and limits generalizability. Cross-sectional design: The design restricts causal inference between occupational noise exposure and tinnitus; longitudinal data would better clarify temporal relationships. Self-reported tinnitus: Reliance on unvalidated self-report introduces potential recall bias and risk of under- or overestimation, particularly for transient symptoms. Gender exclusivity: The sample consisted solely of male generator technicians, reflecting the occupational demographics of informal repair work in Nigeria, but limiting generalizability to female noise-exposed populations or other informal sectors. Uncontrolled confounders: Factors such as recreational noise exposure, smoking, medication use, and mental health status were not explicitly captured due to field constraints, though they may influence tinnitus risk. Geographic limitation: The study was conducted in a single urban Nigerian state, and regional differences in exposure profiles and practices may limit external validity. Audiological equipment constraints: Although testing was performed under controlled conditions, reliance on portable DPOAE devices may reduce sensitivity compared to advanced clinical equipment. Future studies should adopt longitudinal designs, include female technicians, utilize validated tinnitus scales, and control for potential confounders to strengthen causal interpretations.

Finally, this study highlights an urgent public health need, recognizing and addressing tinnitus as a proxy for occupational auditory risk in Nigeria's informal economy sector. With nearly half of generator technicians reporting symptoms, and subclinical cochlear damage detectable via OAE testing, this cohort study points toward scalable interventions, ranging from mobile hearing screening units to hearing protective device advocacy and occupational noise regulation in informal settings.

## Conclusion

This study found that tinnitus affects nearly half of informal generator technicians in urban Kano-Nigeria, with prolonged occupational noise exposure and OAE failure serving as significant predictors of the condition. These findings highlight both the prevalence of cochlear dysfunction and the potential of OAE screening to detect auditory damage earlier than conventional methods. Given the limited health safeguards in informal work environments, integrating hearing protection, noise education, and portable OAE testing into community outreach programs presents a practical and culturally responsive strategy. Expanding auditory health surveillance, especially for youth entering noisy trades, e.g., generator repair settings, could help preserve long-term functional capacity, economic productivity, and social wellbeing. Future longitudinal studies would examine symptom trajectories, broader systemic barriers to auditory health equity across informal labour populations, tinnitus progression, gender differences, and the impact of targeted interventions tailored to informal sector dynamics in low-resource settings.

## Supporting information

**S1 Fig. Smoothed curve illustrating the relationship between age and tinnitus prevalence.** The solid line represents the fitted curve; straight lines denote the 95% confidence interval.
(TIF)

**S2 Fig. Forest plot of adjusted odds ratios (aOR) with 95% confidence intervals for predictors of tinnitus.** Diamond-shaped points indicate estimates; horizontal lines show confidence intervals; the vertical line marks the null value (aOR = 1).
(TIF)

**S1 Data. Raw dataset for Tinnitus study.**
(XLSX)

**S1 Table. List of raw data legend.**
(DOCX)

## Acknowledgments

The authors extend their sincere gratitude to the leadership and members of the Generator Mechanics Union in Kano for their invaluable collaboration throughout the study. Appreciation is also due to the audiology team at the ENT Clinic, Aminu Kano Teaching Hospital, whose technical expertise and support during data collection were instrumental to the success of this research.

## Author contributions

**Conceptualization:** Fatimah Ismail Tsiga-Ahmed, Abdulazeez Ahmed.

**Data curation:** Nafisatu Bello-Muhammad.

**Formal analysis:** Fatimah Ismail Tsiga-Ahmed, Nafisatu Bello-Muhammad, Abdulazeez Ahmed.

**Investigation:** Fatimah Ismail Tsiga-Ahmed.

**Methodology:** Fatimah Ismail Tsiga-Ahmed, Nafisatu Bello-Muhammad, Abdulazeez Ahmed.

**Resources:** Nafisatu Bello-Muhammad.

**Software:** Abdulazeez Ahmed.

**Supervision:** Abdulazeez Ahmed.

**Validation:** Fatimah Ismail Tsiga-Ahmed, Abdulazeez Ahmed.

**Writing – original draft:** Fatimah Ismail Tsiga-Ahmed, Nafisatu Bello-Muhammad, Abdulazeez Ahmed.

**Writing – review & editing:** Fatimah Ismail Tsiga-Ahmed, Nafisatu Bello-Muhammad, Abdulazeez Ahmed.

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
