## [Decision Letter · Decision Letter 0]

10 Sep 2025

PGPH-D-25-02022

Tinnitus and Occupational Noise Exposure among Informal Generator Technicians: Insights from a Nigerian Pilot Study

Dear Dr. Ahmed,

Thank you for submitting your manuscript to PLOS Global Public Health. After careful consideration, we feel that it has merit but does not fully meet PLOS Global Public Health’s publication criteria as it currently stands. Therefore, we invite you to submit a revised version of the manuscript that addresses the points raised during the review process.

Please note that we have only been able to secure a single reviewer to assess your manuscript. We are issuing a decision on your manuscript at this point to prevent further delays in the evaluation of your manuscript. Please be aware that the editor who handles your revised manuscript might find it necessary to invite additional reviewers to assess this work once the revised manuscript is submitted. However, we will aim to proceed on the basis of this single review if possible.

The reviewer has highlighted some major concerns that need to be addressed, particularly surrounding the methodological design and reporting. Please note it is a requirement of publication that methods were conducted to a high standard and are reported in sufficient detail. We ask that you provide a detailed response to the reviewer and update your manuscript accordingly.

We look forward to receiving your revised manuscript.

Kind regards,

Joanna Tindall, PhD

Staff Editor

Journal Requirements:

1. In the online submission form, you indicated that “The datasets generated and analysed during the current study are available from the corresponding author on request.”.

a) In a public repository,

b) Within the manuscript itself, or

c) Uploaded as supplementary information.

2. Please provide separate figure files in .tif or .eps format only and remove any figures embedded in your manuscript file. Please also ensure that all files are under our size limit of 10MB. Please leave the figure captions in the manuscript.

Additional Editor Comments (if provided):

Reviewer #1:

Reviewers' comments:

Reviewer's Responses to Questions

**Comments to the Author**

1. Does this manuscript meet PLOS Global Public Health’s publication criteria?

Reviewer #1: Partly

2. Has the statistical analysis been performed appropriately and rigorously?

Reviewer #1: Yes

3. Have the authors made all data underlying the findings in their manuscript fully available (please refer to the Data Availability Statement at the start of the manuscript PDF file)?

Reviewer #1: Yes

4. Is the manuscript presented in an intelligible fashion and written in standard English?

Reviewer #1: No

Reviewer #1: This study focuses on the problem of tinnitus among informal generator repair workers in Nigeria, and the topic has significant practical significance and public health value. The research method combines questionnaire surveys, pure tone audiometry, and otoacoustic emission (OAE) tests, and the design is comprehensive. The data analysis is rigorous. The results clearly show that the prevalence of tinnitus is high (45.2%), and it is significantly related to working experience of ≥ 10 years and OAE failure, highlighting the potential of OAE in the early detection of subclinical cochlear damage. However, the study has several limitations: the sample size is small and limited to males, which restricts the generalizability of the results; tinnitus is self-reported and no standardized tools were used; no control was made for confounding factors such as recreational noise and mental health; the sensitivity of the portable OAE device may be insufficient; the cross-sectional design cannot infer causal relationships. It is recommended that the authors fully clarify these limitations in the discussion and consider expanding the sample size, using validated tools, including more control variables, and conducting longitudinal studies in the future. Overall, this study provides valuable basic evidence for occupational hearing conservation in resource-poor areas.

**Do you want your identity to be public for this peer review?** For information about this choice, including consent withdrawal, please see our Privacy Policy

Reviewer #1: No

---

## [Decision Letter · Decision Letter 1]

20 Oct 2025

PGPH-D-25-02022R1

Tinnitus and Occupational Noise Exposure among Informal Generator Technicians: Insights from a Nigerian Pilot Study

Dear Dr. Ahmed,

Thank you for submitting your manuscript to PLOS Global Public Health. After careful consideration, we feel that it has merit but does not fully meet PLOS Global Public Health’s publication criteria as it currently stands. Therefore, we invite you to submit a revised version of the manuscript that addresses the points raised during the review process.

The revised manuscript has been assessed by two reviewers who have provided some additional points that should be addressed in your revisions. The comments can be found below, please review these and make the appropriate changed to address any concerns. Please note any suggested citations from Reviewer 2 are not required, please ensure these are relevant to your research before including them.

We look forward to receiving your revised manuscript.

Kind regards,

Emma Campbell, Ph.D

Staff Editor

Journal Requirements:

Additional Editor Comments (if provided):

Reviewers' comments:

Reviewer's Responses to Questions

**Comments to the Author**

Reviewer #1: (No Response)

Reviewer #2: (No Response)

publication criteria?

Reviewer #1: Partly

Reviewer #2: Partly

3. Has the statistical analysis been performed appropriately and rigorously?

Reviewer #1: No

Reviewer #2: No

4. Have the authors made all data underlying the findings in their manuscript fully available (please refer to the Data Availability Statement at the start of the manuscript PDF file)?

Reviewer #1: Yes

Reviewer #2: Yes

5. Is the manuscript presented in an intelligible fashion and written in standard English?

Reviewer #1: (No Response)

Reviewer #2: Yes

Reviewer #1: 1.Emphasize the pilot nature of the study in the abstract and conclusion to avoid overgeneralization.

2.Clarify limitations: self-report tinnitus measure, absence of direct noise level recordings, male-only sample.

3.Strengthen the methods section by briefly justifying why no standardized tinnitus questionnaire was used.

4.Refine the discussion of psychosocial factors (marital status, stress, etc.)—frame these as hypotheses for future study rather than strong conclusions.

Reviewer #2: This manuscript by Tsiga-Ahmed et al. describes the results of a small cohort study of noise-exposed informal generator technicians in Nigeria, reporting the prevalence of tinnitus, audiological features, and sociodemographics of participants. While the authors should be commended for focusing on this important topic in an underserved population that will undoubtedly benefit from occupational hazard intervention/counseling, the article and presentation of results currently do not meet publication standards. But with revision, I think the results are worth publishing and shed light on the substantial problem of tinnitus in a unique population.

MAJOR COMMENTS

- A main deficiency in the article is the presentation of the results in the tables and figures. Notably, there should be a table comprehensively summarizing the demographic/social characteristics of the cohort (as Table 1), as is standard for cohort studies and epidemiological research. The authors are encouraged to review other published papers in this journal for guidance, but summary statistics for all data listed in the 'List of Raw Data' table should be reported for the entire cohort.

- The current tables are repetitive with the Results text and often do not provide additional information or are so short as to be better reported simply in the text (i.e., Table 2, 3). Table 4 appears to be partially redundant with Figure 1 and does not report the proportions of each group reporting tinnitus.

- Three-dimensional bars graphs are not appropriate for publication so Figure 1 should be re-made as a flat stacked bar graph, with Ns and proportions clearly labeled. Forest plots illustrating the results of the regressions would also be helpful.

- Ensure that all factual statements or claims about what is said in the literature are backed up by appropriate citations (e.g., lines 146-148, lines 238-240, etc.)

- Methodological details are scant and need to be expanded as listed below. An example informed consent form and examples of all assessment instruments should be included in the Supplementary Materials given non-standard items were used.

- Was any counseling or education provided to participants? Were they made aware of the study results? Provided with hearing protection?

- The lack of an association with age is very surprising (given that it is a strongly established risk factor for tinnitus, especially in the context of noise exposure) and may be due to a lack of statistical rigor. How do the authors explain this finding? There should be a table tabulating tinnitus prevalence by age groups (e.g., decades) and plot it (age on x, % with tinnitus on y). Then fit a locally smoothed curve (e.g., LOESS) of prevalence vs. age to see shape. What did a logistic regression of age say (Model: logit[P(tinnitus=1)] = β0 + β1·Age). You could report odds ratio per 10-year increase (scale age/10 so β1 is per decade).

SPECIFIC COMMENTS

Abstract:

- The demographics of the cohort need to be better described (for example, age is missing).

Introduction:

- Line 58-59: This wide range needs context from the smallest to largest proportion. Please specify what type of tinnitus you are referring to (i.e., subjective, non-pulsatile, etc.).

- Line 73: What is the typical dB of the generators? Was this measured at any point (or cite the literature)?

- Line 82: Why is this called a pilot study instead of a cohort study?

Methods:

- Please define the type of tinnitus you are assessing (subjective non-pulsatile?) and who administered all testing/instruments along with their expertise. Was testing on-site or at an institution?

- Line 109: There is a statement that the cohort is representative of the field population but how is not stated. Please expand.

- Line 111-112: The informed consent forms are not in the Supplemental Materials and should be included. How was consent obtained from minor (parents)? Depending on the journal requirements, the IRB approval may need to be mentioned in the Methods.

- Line 112-113: It is unclear why people over age 57 years were excluded as presbycusis starts early in noise-exposed populations. A better approach is probably to not exclude by age, but include age as a variable in more detailed analyses to understand its impact.

Line 121: What was the size of the registries randomly sampled from?

Line 125: Table 1 also includes mention of collection of family history of deafness. Those methods and definitions need to be fully described in the Methods section. Was it limited to immediate family? Genetic or acquired deafness? Self-report or diagnosis by physician?

Line 128: What instrument was used to assess tinnitus history? All instruments should be included in the Supplemental Materials.

Line 131: Who administered the PTA and OAE tests? It seems you have the granularity to report more detailed information for the PTA tests (e.g., by frequencies) and I suggest you do so. It would be important to understand any differences in high vs. low frequency hearing loss. Additionally, no results of the otoscopic exams are reported.

Line 138-139: For OAE testing, please define what "fail" versus "pass indicates (what are the parameters?)

Line 141: How were ambient noise levels controlled if performed on-site?

Line 156: List out the variables you are referring to

RESULTS

- Adding subheadings would help with organizing the presentation of results, beginning with the demographics and characteristics of the cohort

- Reporting of the demographic characteristics needs to be improved and more comprehensive, in line with standard epidemiological reporting in studies. There should be a dedicated table for this. For age, please report the Ns across each decade encompassing the cohort/

- The significance of 'marital status' could be a function of age. What was the mean age of the married vs. unmarried participants?

- In Table 1, please add footnotes with the definitions of "audiometric impairment", "OAE failure", and "Family History of Deafness"

- Line 191: Protective is not the right word. OAE performance (pass) is an indicator of better OHC health and function, i.e., lower levels of damage to the sensory cells that could be responsible for generating tinnitus. In this case, worse OAE function is associated with the presence of tinnitus.

- Table 2 is not needed and these findings can simply be reported in the text. If kept, "Other Covariates" need a footnote listing them out.

- Line 198-199: This statement does not seem to be true because there is a large increase (doubling) of the prevalence of tinnitus in participants age 25 y and younger versus those older than 25 y. This further suggests that the impact of age is not being sufficiently assessed

- In Table 4, the proportions of each group reporting tinnitus should be added

DISCUSSION

- Unless there is specific rationale for being labeled a pilot study, I suggest using 'cohort study.' Otherwise, please add the rationale somewhere in the Discussion

- Line 228-230: Prior studies reporting this finding should be mentioned/discussed:

https://www.sciencedirect.com/science/article/pii/S0378595597000439

https://pmc.ncbi.nlm.nih.gov/articles/PMC12346921/

https://pmc.ncbi.nlm.nih.gov/articles/PMC12297068/

https://www.sciencedirect.com/science/article/pii/S0385814609000030

https://www.sciencedirect.com/science/article/pii/S0385814609002077

- Lines 232-234: Regarding the statement about the integration of OAE screening in low-resources settings, isn't this already the case? It is routinely used for newborn screening with portable devices worldwide

- Line 243: The statement about the insignificance of age is not clear from the reported data. There should be a table tabulating tinnitus prevalence by age groups (e.g., decades) and plot it (age on x, % with tinnitus on y). Then fit a locally smoothed curve (e.g., LOESS) of prevalence vs. age to see shape. What did a logistic regression of age say (Model: logit[P(tinnitus=1)] = β0 + β1·Age). Report odds ratio per 10-year increase (scale age/10 so β1 is per decade).

- Lines 247-249: Regarding the significance of marriage, what is the mean age of the married vs. unmarried men? Do the married men work longer hours/have more daily exposure due to supporting a family?

Lines 262: Child labor is indeed a concerning feature of this cohort, so extra care need to be taken in the reporting of the informed consent. Was any counseling or education provided to participants? Were they made aware of the study results? Provided with hearing protection?

Line 276-278: Please provide the rational for why the Tinnitus Functional Index or Tinnitus Handicap Inventory, etc. wasn't used? Lack of a translated version?

**Do you want your identity to be public for this peer review?** For information about this choice, including consent withdrawal, please see our Privacy Policy

Reviewer #1: No

Reviewer #2: No

---

## [Decision Letter · Decision Letter 2]

25 Jan 2026

PGPH-D-25-02022R2

Tinnitus and Occupational Noise Exposure among Informal Generator Technicians in Nigeria: A Pilot Cohort Study

Dear Dr. Ahmed,

Thank you for submitting your manuscript to PLOS Global Public Health. After careful consideration, we feel that it has merit but does not fully meet PLOS Global Public Health’s publication criteria as it currently stands. Therefore, we invite you to submit a revised version of the manuscript that addresses the points raised during the review process.

We look forward to receiving your revised manuscript.

Kind regards,

Helen Howard

Staff Editor

Journal Requirements:

Additional Editor Comments (if provided):

Reviewers' comments:

Reviewer's Responses to Questions

**Comments to the Author**

Reviewer #2: All comments have been addressed

Reviewer #3: (No Response)

publication criteria?

Reviewer #2: Yes

Reviewer #3: Yes

3. Has the statistical analysis been performed appropriately and rigorously?

Reviewer #2: Yes

Reviewer #3: No

4. Have the authors made all data underlying the findings in their manuscript fully available (please refer to the Data Availability Statement at the start of the manuscript PDF file)?

Reviewer #2: Yes

Reviewer #3: Yes

5. Is the manuscript presented in an intelligible fashion and written in standard English?

Reviewer #2: Yes

Reviewer #3: Yes

Reviewer #2: The authors have satisfactorily addressed my comments. Thank you for this important contribution on an under-studied population.

Reviewer #3: Congratulations for an informative manuscript, with conclusions that are largely well founded.

I have two issues which need addressing:

1. line 119: Please provide some more detail about the sample size calculation - for example was it designed to assess the prevalence of tinnitus, or to assess a particular magnitude of association of a categorical or continuous variable with tinnitus? If so, that is the power, effect size and type I error rate considered?

The reference here is indicating that in designing a study to detect a problem, then 60 patients is appropriate. The study in this manuscript is not a study to detect a problem, rather a study examining factors associated with tinnitus - thus it is inappropriate to justify the sample size with that reference. A statistician may be able to assist the author with justification of their sample size.

2. Figure 1/ Table 2 and table 3: please add 95% confidence intervals for the rates - to the numerical estimates and the graph. This helps to show the uncertainty of the estimates.

Typos:

line 219 - extra %

**Do you want your identity to be public for this peer review?** For information about this choice, including consent withdrawal, please see our Privacy Policy

Reviewer #2: No

Reviewer #3: No

---

## [Decision Letter · Decision Letter 3]

11 Feb 2026

Tinnitus and Occupational Noise Exposure among Informal Generator Technicians in Nigeria: A Pilot Cohort Study

PGPH-D-25-02022R3

Dear Ahmed,

We are pleased to inform you that your manuscript 'Tinnitus and Occupational Noise Exposure among Informal Generator Technicians in Nigeria: A Pilot Cohort Study' has been provisionally accepted for publication in PLOS Global Public Health.

Best regards,

Julia Robinson

Executive Editor

Reviewer Comments (if any, and for reference):

Reviewer's Responses to Questions

**Comments to the Author**

Reviewer #3: All comments have been addressed

publication criteria?

Reviewer #3: (No Response)

3. Has the statistical analysis been performed appropriately and rigorously?

Reviewer #3: (No Response)

4. Have the authors made all data underlying the findings in their manuscript fully available (please refer to the Data Availability Statement at the start of the manuscript PDF file)?

Reviewer #3: (No Response)

5. Is the manuscript presented in an intelligible fashion and written in standard English?

Reviewer #3: (No Response)

Reviewer #3: (No Response)

**Do you want your identity to be public for this peer review?** For information about this choice, including consent withdrawal, please see our Privacy Policy

Reviewer #3: No
